# mTOR Signaling Pathway Regulates Sperm Quality in Older Men

**DOI:** 10.3390/cells8060629

**Published:** 2019-06-21

**Authors:** Joana Vieira Silva, Madalena Cabral, Bárbara Regadas Correia, Pedro Carvalho, Mário Sousa, Pedro Fontes Oliveira, Margarida Fardilha

**Affiliations:** 1Laboratory of Signal Transduction, Department of Medical Sciences, Institute of Biomedicine–iBiMED, University of Aveiro, 3810-193 Aveiro, Portugal; joanavieirasilva@ua.pt (J.V.S.); regadas.correia@gmail.com (B.R.C.); pedroacarvalho@ua.pt (P.C.); 2i3S-Instituto de Investigação e Inovação em Saúde, University of Porto, 4200-135 Porto, Portugal; pfobox@gmail.com; 3Unit for Multidisciplinary Research in Biomedicine (UMIB), Institute of Biomedical Sciences Abel Salazar (ICBAS), University of Porto, 4050-313 Porto, Portugal; msousa@icbas.up.pt; 4COGE-Clínica Obstétrica e Ginecológica de Espinho, 4500-057 Espinho, Portugal; madalenaribeirocabral@gmail.com; 5Department of Microscopy, Laboratory of Cell Biology, Institute of Biomedical Sciences Abel Salazar (ICBAS), University of Porto, 4050-313 Porto, Portugal

**Keywords:** aging, sperm quality, signaling proteins, mTORC1, TP53

## Abstract

Understanding how age affects fertility becomes increasingly relevant as couples delay childbearing toward later stages of their lives. While the influence of maternal age on fertility is well established, the impact of paternal age is poorly characterized. Thus, this study aimed to understand the molecular mechanisms responsible for age-dependent decline in spermatozoa quality. To attain it, we evaluated the impact of male age on the activity of signaling proteins in two distinct spermatozoa populations: total spermatozoa fraction and highly motile/viable fraction. In older men, we observed an inhibition of the mechanistic target of rapamycin complex 1 (mTORC1) in the highly viable spermatozoa population. On the contrary, when considering the entire spermatozoa population (including defective/immotile/apoptotic cells) our findings support an active mTORC1 signaling pathway in older men. Additionally, total spermatozoa fractions of older men presented increased levels of apoptotic/stress markers [e.g., cellular tumor antigen p53 (TP53)] and mitogen-activated protein kinases (MAPKs) activity. Moreover, we established that the levels of most signaling proteins analyzed were consistently and significantly altered in men older than 27 years of age. This study was the first to associate the mTOR signaling pathway with the age impact on spermatozoa quality. Additionally, we constructed a network of the sperm proteins associated with male aging, identifying TP53 as a central player in spermatozoa aging.

## 1. Introduction

Infertility is a growing concern in modern societies, especially in industrialized countries where birth rates are declining drastically below replacement levels [1]. Infertility affects approximately 15% of couples attempting to conceive and, in half of the cases, the cause is related to male reproductive issues [1]. Moreover, idiopathic infertility remains the most common type of male infertility [2,3].

The trend in parenthood at an older age is increasingly more pronounced, with the age of first child averaging 30 years in several countries [4]. In addition to intentionally delayed childbearing due to professional activities, increased rates of obesity, sedentary lifestyle, and alcohol and nicotine consumption found in reproductive-aged men and women further perpetuate the problem. While it is well documented that women have a decline in fecundity with age [5], the studies available regarding the effects of age on male fertility have a wider disparity of results. Most studies are mainly focused on basic semen parameters and DNA integrity [6]. Most of these studies reported lower semen volume, sperm concentration, sperm motility, and proportions of sperm of normal morphology in older men; still, some found semen quality to be independent from age or even reported improvements in semen traits [6,7]. Additionally, most studies are limited to a specific study population and are influenced by the female partner age [7,8]. Stone et al. reported that basic seminal parameters start to decline at 34 years of age [9]. Numerous studies described adverse associations between male age and reproductive outcomes. Advanced paternal age was associated with lower pregnancy rates, higher risk of pregnancy loss, and a broad range of developmental, morphologic and neurologic disorders of the newborn, including schizophrenia, autism, X-linked recessive and autosomal dominant disorders, trisomy, and some cancers [6,7,10,11,12]. While the molecular mechanisms are still not understood, aging is associated with an increase in oxidative stress, and a decline in sperm quality was correlated with the excessive generation of reactive oxygen species (ROS) [6].

Since the molecular mechanisms responsible for age-dependent decline in spermatozoa quality are not fully understood, in this study, we evaluated the impact of aging on human spermatozoa signaling pathways, including the mitogen-activated protein kinases (MAPKs), the mechanistic target of rapamycin (mTOR), and the apoptosis signaling pathways. Given the heterogeneous nature of the ejaculate in terms of sperm integrity/functionality, two distinct spermatozoa populations were analyzed: total spermatozoa fractions and highly motile/viable fractions. Additionally, we determined whether age thresholds for the signaling proteins examined existed. An interaction network of the sperm proteins associated with male age was constructed and analyzed.

## 2. Material and Methods

### 2.1. Ethical Approval

This study was approved by the Ethics and Internal Review Board (Process number: 36/AO) of the Hospital Infante D. Pedro E.P.E., Aveiro, Portugal and was conducted in accordance with the ethical standards of the Helsinki Declaration. All participants received clear written instructions concerning sample collection, study design, and signed informed consent, allowing the samples to be used for scientific purposes. No monetary compensation was applied. All data were anonymized.

### 2.2. Study Design

Given the heterogeneous nature of the ejaculate in terms of sperm integrity/functionality, two distinct populations were analyzed: total spermatozoa fractions and highly motile/viable fractions.

A total of 93 human semen samples were included in this study from volunteers randomly assigned to one of the two cohorts. The dataset included both volunteers recruited at the University of Aveiro (Aveiro, Portugal) and men providing semen samples for routine analysis at Hospital Infante D. Pedro, E.P.E. (Aveiro, Portugal) and at COGE Obstetric and Gynecological Clinic of Espinho (Espinho, Portugal). Exclusion criteria comprised varicocele, genital tract infection, cryptorchidism and testicular trauma, systemic illness, and use of any medications.

### 2.3. Semen Analysis and Processing

Semen samples were obtained by masturbation into a sterile container and delivered for basic semen analysis within 30 min. All samples were analyzed according to the World Health Organization criteria by experienced technicians [13].

Semen samples were processed using either a simple wash method or a density gradient centrifugation followed by swim-up. In the simple wash method, after complete liquefaction, 1× phosphate-buffered saline (PBS) was added to the ejaculate and centrifuged (5 min at 500× *g*) twice to remove the seminal plasma. Spermatozoa were subsequently used for the antibody array. It is worth noting that samples were thoroughly scanned for the presence of somatic cells (including leucocytes) and only samples that did not contain somatic cells were subjected to this washing method. This analysis was performed according to the World Health Organization guidelines [13] by examining a stained smear at 1000× magnification.

For the density gradient centrifugation, semen was centrifuged with gradients SupraSperm (company, Origio, Denmark) at 55% and 80%, for 20 min at 300× *g*. The supernatant was carefully aspirated, and the sperm pellet was resuspended and washed with Sperm Preparation Medium (CooperSurgical Fertility & Genomic Solutions, Malov, Denmark) for 10 min at 300× *g*. Sperm Preparation Medium (company) was carefully added above the pellet and incubated at a 45° angle for 1 h for swim-up in a vertical rack in a 37 °C incubator. Spermatozoa were subsequently used for the antibody array.

For the antibody array, spermatozoa were incubated with 1× PathScan Sandwich ELISA Lysis Buffer (Cell Signaling Technology, Danvers, MA, USA) supplemented with 1 mM phenylmethanesulfonyl fluoride (PMSF) for 5 min on ice. Then, the lysed cells were centrifuged at 14,000× *g* at 4 °C for 10 min, and the supernatant was transferred to a new tube and stored at −80 °C.

### 2.4. Protein Quantification

Protein concentration was measured using a bicinchoninic acid assay (Pierce BCA Protein Assay Kit, Thermo Fisher Scientific, Waltham, MA, USA), and final absorbance was measured at 562 nm in a microplate reader (Infinite^®^ 200 PRO series, Tecan Trading AG, Mannedorf, Switzerland).

### 2.5. Antibody Array

Antibody-based arrays were carried out using the PathScan^®^ Intracellular Signaling Array Kit (#7744, Cell Signaling Technology) to determine the expression patterns of 18 well-characterized signaling molecules when phosphorylated or cleaved, in 93 semen samples. Sixty-three spermatozoa samples were analyzed using extracts prepared by the simple wash method, and 30 samples were prepared with gradient density centrifugation followed by swim-up.

The Intracellular Signaling Array Kit allows simultaneously detecting multiple molecules from a wide range of key signaling pathways for mammalian cells including the MAPK, mTOR, and apoptosis signaling pathways.

Each cell extract was diluted to 250 μg/mL and applied to its own multiplexed array according to the manufacturer’s instructions. Fluorescence readouts from the arrays were captured digitally using LI-COR^®^ Biosciences Odyssey^®^ imaging system (LI-COR^®^ Biosciences, City, NE, USA). Pixel intensity was quantified using Odyssey software. The intensity from the negative control (without an antibody captured onto the array) within each array was subtracted from all signals, and all data from each array were normalized to the internal positive control within each array.

### 2.6. Network and Bioinformatics Analysis

A literature search was conducted using the PubMed database to identify human sperm proteomic studies, including proteins differentially expressed between young adults and aged men. By merging data from the available papers, a list of 7622 different proteins present in human spermatozoa was created (30 April 2018).

The testis/sperm-enriched/specific proteins were extracted from the Human Protein Atlas (HPA) dataset [14], and a list of 2237 different proteins was extracted (27 February 2018).

The Human Integrated Protein-Protein Interaction rEference (HIPPIE) database was used for retrieving human protein–protein interaction (PPI) data (23 April 2018). This database is regularly updated by incorporating interaction data from major expert-curated experimental PPI databases [15]. The protein network was built using Cytoscape V3.5.1 software (Institute for Systems Biology, Seattle, WA, USA) [16].

Functional enrichment analysis of Gene Ontology categories was performed using the Panther overrepresentation test. The overall set of human protein-coding genes was used as reference set, and only enriched functions with *p* < 0.05 (including Bonferroni correction) were considered.

Protein identities associated with defects in male fertility or a functional/morphological defect in the male reproductive system were obtained from the DisGeNET database [17] and the Mouse Genome Informatics (MGI) database [18].

### 2.7. Statistical Analysis

We performed an observational study of a cohort of 93 healthy men. Initially we conducted an exploratory data analysis (EDA) using graphical techniques and quantitative analysis to characterize the sample, and to detect possible extreme outliers and measurement errors. Quantitative variables were firstly checked for normality using Shapiro–Wilk tests together with an inspection of their graphical representation.

We evaluated the impact of male age on the activity of signaling proteins in two distinct spermatozoa populations: total spermatozoa fraction and highly motile/viable fraction. Correlations between age and the expression of signaling proteins were evaluated using Pearson’s correlation coefficient. In the total spermatozoa fraction population, for the variables with at least moderate correlation, the power of age as a predictor of change of the molecular parameters was quantified by the area under the Receiver Operating Characteristic (ROC) curve (AUC) (assuming a parameter change when greater than or equal to mean ± SD). When the area under the ROC curve was significantly higher 0.7, we calculated the cutoff (age) applying Youden’s index. This technique did not allow determining age thresholds in the highly motile/viable spermatozoa fraction, since the area under the curve was not significant. After this exploratory analysis, to investigate the existence of differences in molecular parameters after 27 years (cutoff previously found), we conducted Mann–Whitney U tests. The assumptions of the statistical techniques used were validated. The statistical analysis was performed using IBM-SPSS version 21 for Windows (IBM Corp, Armonk, NY, USA). A significance level of 0.05 was adopted.

## 3. Results

### 3.1. Age Impacts the Activity of Several Signaling Proteins

The levels of 18 signaling proteins in distinct status of activation were determined in human spermatozoa using an antibody array which allowed analyzing a wide range of key signaling pathways for mammalian cells including the MAPK, mTOR, and apoptosis signaling pathways (Appendix A).

A total of 63 spermatozoa samples were analyzed using extracts prepared by a simple wash method, which allowed the recovery of all spermatozoa populations, rather than a sperm subpopulation (Appendix A). The volunteers’ ages ranged from 18 to 47 years (mean ± SD = 26 ± 7), and only seven were smokers. It is worth noting that the 63 men included in this dataset did not undergo assisted reproduction technology (ART) due to male factor. The results indicated that the levels of several phosphoproteins in human spermatozoa were positively correlated with age: cellular tumor antigen p53 (TP53) (S15), Bcl2-associated agonist of cell death (BAD) (S112), MAPK14/11/12/13 (T180/Y182), MTOR (S2448), Proline-rich AKT1 substrate 1 (AKT1S1) (T246), heat shock protein beta-1 (HSPB1) (S78), RAC-alpha serine/threonine-protein kinase (AKT) (S473), Glycogen synthase kinase-3 beta (GSK3B) (S9), MAPK8 (T183/Y185), and 40S ribosomal protein S6 (RPS6) (S235/236). The cleavage and consequent activation of Caspase-3 (CASP3) (D175) was also positively correlated with age (Table 1; Appendix A).

To determine whether age thresholds existed, the area under the ROC curve (AUC) was analyzed for each molecular parameter. It was established that the levels of most signaling proteins were analyzed consistently and significantly increased after 27 years of age (AUC greater than 0.726, sensibility greater than 88%, and specificity greater than 69%) (Appendix A). In fact, the levels of all signaling proteins found to be associated with age were significantly increased in the population aged 27 or more (Appendix A), most of which were associated with the mechanistic target of rapamycin (MTOR) signaling cascade (Figure 1).

To evaluate the levels of the signaling molecules in highly motile, morphologically normal, viable spermatozoa, a total of 30 semen samples were prepared with gradient density centrifugation followed by swim-up (Appendix A). It is worth noting that, from the 30 men, only one resorted to ART due to male factor (no pregnancy was achieved), and only five were smokers. Ages ranged from 18 to 46 years (mean ± SD = 37 ± 7). In the motile/viable spermatozoa population, the levels of two phosphoproteins were negatively correlated with age: AKT1S1 (T246) and RPS6KB1 (T389) (Table 1; Appendix A). Given the relatively small sample size in this group (*N* = 30), it was not possible to determine an age threshold. Nevertheless, when considering the age threshold identified previously and despite only three volunteers being below that age cutoff, the levels of inhibited AKT1S1 and activated MAPK8 (both associated with the MTOR pathway) were significantly decreased in men aged over 27 years old (Figure 1; Appendix A).

### 3.2. Network of Sperm Proteins Associated with Male Age

A protein–protein interaction (PPI) network was constructed based on information gathered from the proteins identified in this study (green nodes, Table 1) and from proteins identified from the literature search (pink and blue nodes, Appendix A) whose activity or expression, respectively, was associated with male age (Figure 2).

The total number of proteins in the PPI network was *N* = 39 and the total number of interactions between them was L = 68. Nine proteins from the dataset had no PPI data available (isolated nodes). In Figure 2, node size is proportional to the number of interactions of each node (k, degree). TP53 (k = 15), AKT1 (k = 13), MTOR (k = 10), GSK3B (k = 8), MAPK8 (k = 8), 14-3-3 protein theta (YWHAQ) (k = 7), and myosin-9 (MYH9) (k = 6) were the proteins with a higher degree of connectivity in the network. The average number of neighbors in the network was 4.533. Proteins identified in this study (green nodes) and proteins gathered from the literature search (pink and blue nodes) were highly interconnected. The mean clustering coefficient was *C* = 0.311. The clustering coefficient characterizes how the nearest neighboring nodes of a node are connected to each other. If all of them are tightly connected to each other, then they form a clique and the clustering coefficient is 1. For a sparse random uncorrelated network of finite size *N*, this coefficient is close to zero. Note that the value *C* = 0.311 is about 16 times larger than the clustering coefficient expected for a sparse random uncorrelated network (in the latter case, *C* = <*q*>/*N*~0.02). When analyzing only the proteins identified by Liu and colleagues [19] as differentially expressed in aged men (blue and pink nodes) and the interactions between them, the clustering coefficient of the network was *C* = 0.00. Among the proteins identified by Liu and colleagues [19], the enrichment analysis revealed that the only biological process category displaying *p* < 0.05 was protein folding (heat shock-related 70 kDa protein 2 (HSPA2), T-complex protein 1 subunit gamma (CCT3), and T-complex protein 1 subunit eta (CCT7); *p* = 2.89 × 10^−2^), with a fold enrichment of 31.98 (expected = 0.09). When considering all the proteins in the network, the significant biological processes were protein folding, MAPK cascade, phosphate-containing compound metabolic process, and response to stimulus (Appendix A).

The PPI network was then expanded by adding the interactors annotated in the Gene Ontology category “aging” (Appendix A). In total, 286 proteins were annotated in Homo sapiens in this category and, from those, 122 were interactors of the proteins in Figure 2. The total number of proteins in the network was *N* = 161 and the total number of interactions between them was L = 446. The mean clustering coefficient of the network was *C* = 0.225 and the average number of neighbors 5.682, with only four isolated nodes. The proteins with a higher degree of connectivity in this network were TP53 (k = 73), HSPA4 (k = 37), AKT1 (k = 36), YWHAQ (k = 28), MAPK8 (k = 27), GSK3B (k = 24), and MYH9 (k = 22). The enrichment analysis revealed that the most significant biological processes were response to stress, apoptotic process, MAPK cascade, DNA repair, regulation of cell cycle, and protein folding (Appendix A). Ninety-one proteins from the network were previously reported in human spermatozoa proteomes and eight are testis/sperm-enriched/specific proteins.

## 4. Discussion

Several studies investigated the age-based impact on semen parameters, such as semen volume, sperm concentration, morphology, motility, and DNA fragmentation [6,7,8], but the impact of paternal age on human spermatozoa signaling pathways remains unknown.

The heterogeneous nature of the ejaculate in terms of sperm integrity/functionality makes it crucial to characterize distinct populations. To that end, total spermatozoa and highly motile/viable fractions were analyzed to determine the association of male age and the activity patterns of key signaling proteins in human spermatozoa.

In this study, the activity of 12 proteins in spermatozoa was correlated with male age. From those, half were main components of the mTORC1 signaling pathway (Table 1, Figure 3).

Mechanistic target of rapamycin (mTOR) functions in two distinct complexes, mTORC1 and mTORC2 [20]. mTORC1 has five components: MTOR, which is the catalytic subunit of the complex; RAPTOR (regulatory-associated protein of MTOR); MLST8 (target of rapamycin complex subunit LST8); and two inhibitory subunits AKT1S1 (proline-rich AKT1 substrate 1, also known as PRAS40) and DEPTOR (DEP domain containing mTOR interacting protein). RPS6KB1 (ribosomal protein S6 kinase beta-1, also known as p70S6 kinase) is a major substrate of mTORC1. Phosphorylation of AKT1S1 by AKT1 at T246 releases AKT1S1 from the mTORC1 complex leading to the binding of phosphorylated AKT1S1 to cytoplasmic docking protein 14-3-3 (YWHA), thereby abolishing the inhibitory effect of AKT1S1 on mTORC1 signaling [20].

In this study, the levels of inhibited AKT1S1 and activated RPS6KB1 were reduced in highly motile/viable spermatozoa of older men, which reflects an inhibition of mTORC1 signaling in this spermatozoa population (Figure 1 and Figure 3). Conversely, in the total spermatozoa fraction of older men, the levels of phosphorylated AKT1, AKT1S1, mTOR, and RPS6 were increased, reflecting an activation of the mTORC1 signaling (Figure 1 and Figure 3). Additionally, we showed that activated MAPK8 is increased in the total spermatozoa fraction and decreased in the highly motile/viable spermatozoa, which is consistent with mTORC1 activation and inhibition, respectively (Figure 3). MAPK8 was described to phosphorylate RAPTOR in stress conditions [21]. RAPTOR phosphorylation normally promotes mTORC1 activity and signaling to downstream substrates [22].

Although there is a consensus that mTOR signaling plays a key role in mammalian aging, the mechanism through which this occurs is still unclear [20,23,24]. While some studies suggested that mTORC1 substrate phosphorylation increases with age, others observed that, in some tissues, mTORC1 signaling decreases with age [23]. A possibility is that inhibition of mTORC1 in the highly viable spermatozoa fraction slows aging by increasing autophagy, which helps clear damaged proteins and organelles such as mitochondria, preventing damage to the cells. In fact, it was previously shown that mTOR signaling inhibition leads to activation of autophagy in rat testes as a self-protective mechanism of the cell in response to external stress [25]. Also, autophagy activation in spermatozoa induced a significant increase in spermatozoa motility [26], which is in accordance with our findings. On the contrary, when considering the entire spermatozoa population (including immotile, apoptotic, and necrotic cells), all our findings support an active mTORC1 signaling pathway. The phosphorylation of the autophagy machinery components by the mTORC1 complex compromises its function, which limits the selective elimination of damaged and dangerous components to the cell, reducing longevity. Reduced autophagy favors the accumulation of oxidized proteins or damaged mitochondria that produce high levels of ROS, which in turn can explain the decline in spermatozoa motility, normal morphology, and unfragmented cells observed in older men. In fact, autophagy inhibition was previously shown to decrease spermatozoa motility and increased caspase 3 activation [26], which we also observed in the total spermatozoa population of older men (Table 1). The role of mTOR signaling pathway in spermatozoa is still unknown and, to our knowledge, this is the first time that mTOR signaling was associated with aging in sperm cells. Ultimately, the importance of mTORC1 signaling in spermatozoa aging likely reflects its unique capacity to regulate a variety of key cellular functions, including autophagy and cell motility. Autophagy-related proteins were only recently shown to be functionally active in human spermatozoa [26]. In testis, RPS6KB1 activity (a major substrate of mTORC1) was linked to the actin cytoskeleton and, consequently, to cell motility [27].

When analyzing the total spermatozoa fractions, rather than a sperm subpopulation, it is important to consider that distinct spermatozoa subpopulations will be present, including cells under adaptive responses (e.g., activation of damage repair mechanisms), as well as apoptotic and necrotic cells. The increased levels of activated TP53 (S15) and CASP3 (D175) in the total spermatozoa fraction of older men, but not in the highly viable fraction, may reflect the contribution of the defective spermatozoa in the activation of these apoptotic markers. CASP3 activity is associated with teratozoospermia and asthenozoospermia, suggesting that nuclear, mitochondrial, and cytoskeletal abnormalities induce CASP3 activation [28]. CASP3 was also shown to be increased in sperm from men with diabetes type I [29]. Serine 15 is the primary target of the DNA damage response on the TP53 protein. TP53 was previously reported to be upregulated in sperm from men with varicocele [30] and DNA fragmentation [31].

In the total spermatozoa fractions of older men, we also observed an activation of the mitogen-activated protein kinases (MAPKs). Several findings support the participation of the MAPK pathway in germ cell apoptosis [32,33]. In mature spermatozoa, this pathway was associated with capacitation and acrosome reaction [32,34]; however, due to conflicting findings, the role of MAPKs remains uncertain. The MAPK pathway is responsible for the phosphorylation of the heat shock protein HSPB1 (also known as HSP27). In fact, in the total spermatozoa fraction of older men, we observed increased levels of phosphorylated HSPB1. Increased HSPB1 expression and phosphorylation are mostly associated with response to different stress stimuli and actin organization [35]. In spermatozoa, increased levels of HSPB1 were previously associated with poor blastocyst development [36] and sperm DNA fragmentation [37].

When analyzing the PPI network of the proteins identified in this study whose activity is correlated with age and the data gathered from the literature search about the proteins that are up and downregulated in older men, the proteins with a higher degree of connectivity were TP53, AKT1, and MTOR. This further supports a key role for mTOR signaling in the spermatozoa aging process and pinpoints a relevant role for TP53 in this process (Figure 3). TP53 has the potential to be a marker for sperm quality in older men since, in addition to being a central hub in the network of sperm proteins associated with age, it presents, among the proteins analyzed, the highest correlation with male age.

In the expanded PPI network, constructed by adding the interactors annotated in the Gene Ontology category “aging” (Appendix A), the most significant biological processes were response to stress, apoptotic process, and MAPK cascade, further pinpointing the potential role of these pathways in the spermatozoa aging process.

It was recently established that semen parameters do not change before 34 years old [9]. Other age thresholds (i.e., 45 and 55 years old) were also reported for several basic semen parameters [38,39,40]. Our analysis revealed that a consistent increase in the levels of most signaling proteins analyzed in total spermatozoa fractions appeared as soon as 27 years of age (Appendix A). This threshold occurs prior to the ones established for basic semen parameters, which suggests that alterations at the molecular level may occur earlier. Rising levels of ROS in semen were reported in men over 40 years old, and ROS were suggested as the instrumental factor for incremental sperm defects with aging [41]. Our analysis revealed alterations at the molecular level occurring before this age threshold, which may indicate other possible mechanisms responsible for the age-dependent decline in sperm quality.

The power of the study to determine age thresholds for the molecular parameters in the highly motile/viable spermatozoa population was limited by the relatively small sample size in this group (*N* = 30). Additionally, the participant age range was limited (18 to 47 years of age) and its distribution differed in both cohorts. We cannot exclude that our findings are due to unmeasured confounders, including lifestyle factors such as alcohol consumption, diet, exercise, and stress. Finally, further studies concerning the total abundance of the sperm proteins identified as associated with male age need to be performed.

This study was the first to associate the mTOR signaling pathway in human spermatozoa with men’s age and to recognize the distinct activation status of this pathway in different spermatozoa populations (Figure 1 and Figure 3). Also, in older men, the total spermatozoa population presented increased activity of apoptotic/stress markers (such TP53) and the MAPK pathway. Additionally, an age threshold for changes in certain spermatozoa signaling parameters was established. Finally, a network of the sperm proteins associated with male age was constructed and analyzed, which allowed supporting a key role for the mTOR signaling in the spermatozoa aging process and which allowed recognizing TP53 as a central player in this process (Figure 2).

Disclosing these molecular signatures of spermatozoa associated with aging is a further step in the development of therapeutic intervention to counteract male subfertility/infertility associated with male senescence.

## Figures and Tables

**Figure 1 cells-08-00629-f001:**
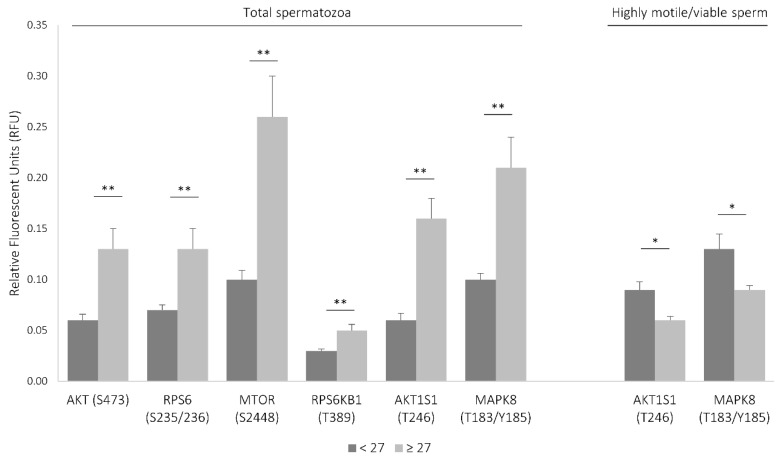
Signaling proteins associated with the mechanistic target of rapamycin (MTOR) pathway differentially activated between groups aged ≤27 and aged >27. Results are expressed as mean ± SEM. The difference between groups was assessed by Mann–Whitney U test. * Difference is significant at the 0.05 level (two-tailed); ** difference is significant at the 0.01 level (two-tailed).

**Figure 2 cells-08-00629-f002:**
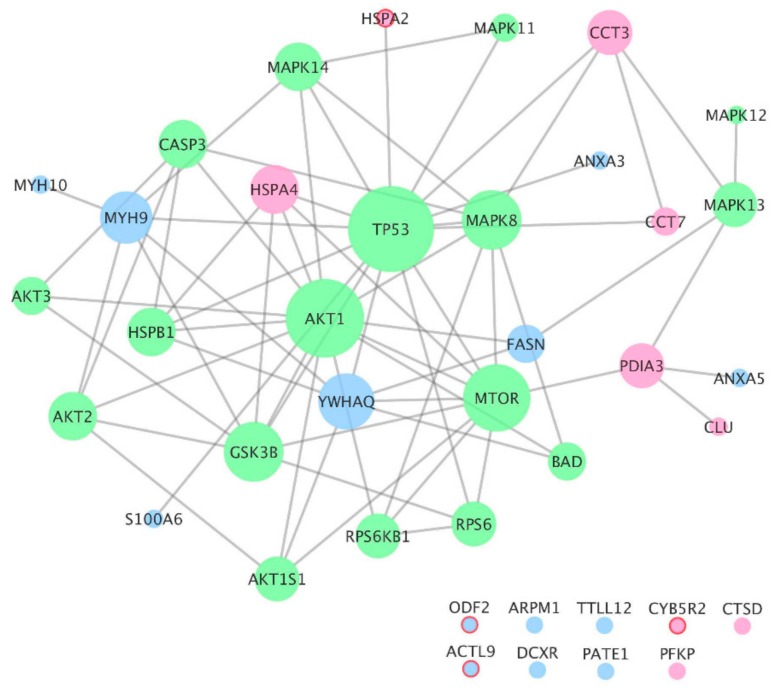
Network of the sperm proteins differentially associated with male age. Green nodes represent the proteins identified in this study whose activity is correlated with age. Pink and blue nodes represent the proteins gathered from the literature search that are upregulated or downregulated in older men, respectively. Red outline represents testis-enriched/specific proteins. Node sizes represent the relative degree of the nodes.

**Figure 3 cells-08-00629-f003:**
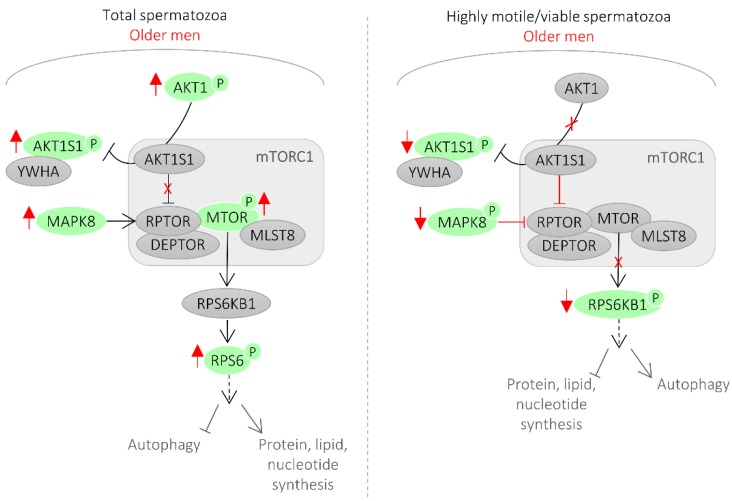
mTOR signaling in spermatozoa from older men. In older men, we observed an inhibition of mTORC1 in the highly viable spermatozoa population. On the contrary, when considering the entire spermatozoa population (including defective, immotile, apoptotic cells), all our findings support an active mTORC1 signaling pathway. Red reflects the findings of this study.

**Table 1 cells-08-00629-t001:** Pearson’s correlation coefficients between age and the activity patterns of signaling proteins in human spermatozoa.

Uniprot ID	Abbreviation		Protein Name	Phosphorylation/Cleavage, Residue, Status	Pearson’s Correlation Coefficient
	Formerly	Presently			
Total Spermatozoa Populations					
P04637	p53	TP53	Cellular tumor antigen p53	Phospho S15 Activation	0.451 **
Q92934	Bad	BAD	Bcl2-associated agonist of cell death	Phospho S112 Inhibition	0.434 **
Q16539	p38	MAPK 11/12/13/14	Mitogen-activated protein kinase 14/11/12/13	Phospho T180/Y182 Activation	0.430 **
Q15759					
P53778					
O15264					
P42345	mTOR	MTOR	Serine/threonine-protein kinase mTOR	Phospho S2448 Activation	0.409 **
Q96B36	PRAS40	AKT1S1	Proline-rich AKT1 substrate 1	Phospho T246 Inhibition	0.406 **
P04792	HSP27	HSPB1	Heat shock protein beta-1	Phospho S78 Activation	0.373 **
P31749	Akt	AKT	RAC-alpha serine/threonine-protein kinase	Phospho S473 Activation	0.365 **
P42574	Caspase-3	CASP3	Caspase-3	Cleavage D175 Activation	0.362 **
P49841	GSK-3β	GSK3B	Glycogen synthase kinase-3 beta	Phospho S9 Inhibition	0.354 **
P45983	SAPK/JNK	MAPK8	Mitogen-activated protein kinase 8	Phospho T183/Y185 Activation	0.335 **
P62753	S6 ribosomal protein	RPS6	40S ribosomal protein S6	Phospho S235/236 Activation	0.317 **
**Highly Motile/Viable Spermatozoa**					
Q96B36	PRAS40	AKT1S1	Proline-rich AKT1 substrate 1	Phospho T246 Inhibition	−0.523 **
P23443	p70 S6 kinase	RPS6KB1	Ribosomal protein S6 kinase beta	Phospho T389 Activation	−0.382 *

* Correlation is significant at the 0.05 level (two-tailed); ** correlation is significant at the 0.01 level (two-tailed).

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
