# Peer review of "mTOR Signaling Pathway Regulates Sperm Quality in Older Men"

_cells, 2019, doi:10.3390/cells8060629_

Round 1

Reviewer 1 Report

This is an observational study on molecular mechanisms responsible for age-dependent decline in spermatozoa quality. Authors tried to associate the mTOR signaling pathway with the age impact on spermatozoa quality and to construct a network of the sperm proteins associated with male aging.

This is  a well conducted study, with some points needing some revision before publication.

The abstract section needs to have a better synchronization between sections, that is methodology, findings and conclusions. Results might look better, if numbers could be added.

In the material and methods section, more detailed description of the includion and exlusion criteria might be needed.

There must be a distinction, for the non expert reader to understand better, the link between the methods and the three signaling patways examined (MAPK, mTOR and apoptosis signaling pathways).

The same observation stands for the rest of the manusript.

The discusiion section should be more structured. Direct comparison with similar studies in the literature is needed. As authors correctly recognise and report the limitations of their study, their conclusion might be milder, to accurately reflect the real findings.

Author Response

We thank the reviewer for the constructive and relevant comments. We have answered all the questions and made the suggested alterations in the manuscript (track changes).

This is a well conducted study, with some points needing some revision before publication.

The abstract section needs to have a better synchronization between sections, that is methodology, findings and conclusions. Results might look better, if numbers could be added.

Authors: The abstract was changed according to the reviewer comment. However, due to abstract word limit, numbers cannot be added.

In the material and methods section, more detailed description of the includion and exlusion criteria might be needed.

Authors: Authors appreciate the observation and a “study design” section was added, with a more detailed description of the inclusion/exclusion criteria.

There must be a distinction, for the non expert reader to understand better, the link between the methods and the three signaling patways examined (MAPK, mTOR and apoptosis signaling pathways).

The same observation stands for the rest of the manusript.

Authors: This aspect was clarified in the final paragraph of the introduction, beginning of the results (section 3.1.) and beginning of the discussion.

The discussion section should be more structured. Direct comparison with similar studies in the literature is needed. As authors correctly recognise and report the limitations of their study, their conclusion might be milder, to accurately reflect the real findings.

Authors: Authors agree with the reviewer and the discussion was reformulated, including citation to additional studies.

Reviewer 2 Report

Dear authors,

I have read your manuscript entitled: “mTOR signaling pathway regulates sperm quality in aged men” with great of interested. From my point of view these types of methodological approaches are recently highly necessary in this research field. As authors have also mentioned, there are many studies related to the men`s factor and basic sperm parameters. However, studies which are going behind the conventional sperm functional/qualitative parameters are lacking.

Your study is of high quality, nevertheless I have some suggestions and comments which I am going to write in the common order.

Abstract

ln 26 – why TP53 is in brackets and MAPKs not? As this is result of your study, I suggest writing all of markers related to mentioned cell events.

lns 29 – 31 – I recommend to split the final sentence to more parts.

I suggest to authors to change word “imperative” in abstract and in another parts of manuscript.

Introduction

lns 44 – 45 – In sentence: “Most studies are mainly focused….” I suggest to authors adding of another citation.

Generally, in this part of manuscript I miss information about signaling pathway which manuscript deals with. I strongly recommend to author to add paragraph about that.

Material and methods

Authors stated that all samples were analysed according to WHO manual. Were basic parameters analysed also after DGC and SU or after simple washing? How did authors decide which samples will be simple washed and which will be processed by DGC + SU? Did authors try to perform any trial experiments with antibody array of sperm population captured in gradient column or which do not swim-up?

lns 84 – 86 – how were samples analysed for presence of somatic cells? I recommend put it into this part.

lns 83, 88, 90 – usually the RCF is written as number x g. So I recommend to change it.

part 2.5. – I strongly recommend to better describe more precisely how many samples per group were analysed. Moreover, addition the experimental design will be very nice.

ln 96 – how the transferred supernatant were stored till analysis?

ln 108 – dilution was based also on manufacturer recommendation?  

lns 111 – 113 – How negative and positive were prepared? I recommend putting it into the text.

ln 127 – delete word “value” after p  

Results

lns 152 – 153 – I recommend removing the last sentence. It belongs to part Material and methods. Moreover, there is already stated, thus here is redundant.  

ln 158 – I suggest to authors check use of the word “resort”  

Table 1 – I suggest modifying heading of table

Discussion

lns 232 – 233 – I recommend adding some citations to the first half of the sentence

regarding the Figure. 3 I suggest removing it from the Discussion. Moreover, I suggest depicting better where is it located in cell

lns 238 – 240 – From my point of view the sentence: “To understand the impact…” does not make a sense. How total population of spermatozoa or their certain subpopulation might express impact of age?

lns 248 – 254 – In this paragraph authors did not present or discuss their results so there I citations are missing.   

ln 254 – last sentence belongs to legend of figure, not to the discussion.

lns 294 – 299 – I strongly recommended to split this very long sentence.

ln 328 – I suggest checking the proper writing of citation in text

ln 329 – which age thresholds?

I recommend merging the last paragraph of the Discussion with conclusion.

Thank you for this work.
My best regards.

Author Response

Authors: We thank the reviewer for the constructive and relevant comments. We have answered all the questions and made the suggested alterations in the manuscript (track changes).

Your study is of high quality, nevertheless I have some suggestions and comments which I am going to write in the common order.

Abstract

ln 26 – why TP53 is in brackets and MAPKs not? As this is result of your study, I suggest writing all of markers related to mentioned cell events.

Authors: TP53 is an example of an apoptotic/stress marker; in addition to alterations in these markers, MAPKs activity, which are involved in a wide range of cellular processes, are also increased. Due to space restrictions in the abstract it was not possible to add all the signaling proteins related to the mentioned cell events. Since TP53 presents the higher correlation with male age and it is also the central hub in the network of sperm proteins associated with age, the authors thought more focus should be given to this protein.

lns 29 – 31 – I recommend to split the final sentence to more parts.

Authors: Authors appreciate the observation and this was modified.

I suggest to authors to change word “imperative” in abstract and in another parts of manuscript.

Authors: This aspect was altered.

Introduction

lns 44 – 45 – In sentence: “Most studies are mainly focused….” I suggest to authors adding of another citation.

Authors: Additional citation were included.

Generally, in this part of manuscript I miss information about signaling pathway which manuscript deals with. I strongly recommend to author to add paragraph about that.

Authors:  As suggested, information was added in the last paragraph of the introduction mentioning the signaling pathways analyzed in this study.

Material and methods

Authors stated that all samples were analysed according to WHO manual. Were basic parameters analysed also after DGC and SU or after simple washing? How did authors decide which samples will be simple washed and which will be processed by DGC + SU? Did authors try to perform any trial experiments with antibody array of sperm population captured in gradient column or which do not swim-up?

Authors: Due to restrictions related with sample amount it was not possible to perform both washing procedures in the same sample. Thus, volunteers were randomly assigned to one of the two cohorts (“study design” section was added to the methods). The study inclusion/exclusion criteria were the same for both cohorts. More detailed information concerning the inclusion/exclusion criteria was also added to the methods (section 2.2.). Samples were also analyzed microscopically after the washing procedures.

lns 84 – 86 – how were samples analysed for presence of somatic cells? I recommend put it into this part.

Authors: This information was added.

lns 83, 88, 90 – usually the RCF is written as number x g. So I recommend to change it.

Authors: As suggested, this was corrected.

part 2.5. – I strongly recommend to better describe more precisely how many samples per group were analysed. Moreover, addition the experimental design will be very nice.

Authors: These are indeed very relevant observations. A “Study design” section was added. Also, a description of the study groups was added to section 2.5.

ln 96 – how the transferred supernatant were stored till analysis?

Authors: The information was added.

ln 108 – dilution was based also on manufacturer recommendation? 

Authors: Yes

lns 111 – 113 – How negative and positive were prepared? I recommend putting it into the text.

Authors: The information was added.

ln 127 – delete word “value” after p 

Authors: The word was eliminated.

Results

lns 152 – 153 – I recommend removing the last sentence. It belongs to part Material and methods. Moreover, there is already stated, thus here is redundant. 

Authors: The sentence was eliminated.

ln 158 – I suggest to authors check use of the word “resort”

Authors:  The word was changed.

Table 1 – I suggest modifying heading of table

Authors: The heading was modified.

Discussion

lns 232 – 233 – I recommend adding some citations to the first half of the sentence

Authors: As pertinently suggested, citations were added.

lns 238 – 240 – From my point of view the sentence: “To understand the impact…” does not make a sense. How total population of spermatozoa or their certain subpopulation might express impact of age?

Authors: Authors agree with the reviewer and the sentence was reformulated.

lns 248 – 254 – In this paragraph authors did not present or discuss their results so there I citations are missing.  

Authors: As pertinently suggested, references were added.

ln 254 – last sentence belongs to legend of figure, not to the discussion.

Authors: The sentence was moved to the figure legend.

lns 294 – 299 – I strongly recommended to split this very long sentence.

Authors: As pertinently suggested, this was altered.

ln 328 – I suggest checking the proper writing of citation in text

Authors: As suggested, this was corrected.

ln 329 – which age thresholds?

Authors: The information was added.

I recommend merging the last paragraph of the Discussion with conclusion.

Authors: The conclusion section was merged into the discussion.